# A novel role for lipid droplets in the organismal antibacterial response

**Preetha Anand[1†], Silvia Cermelli[1†], Zhihuan Li[2], Adam Kassan[3], Marta Bosch[3], Robilyn Sigua[1], Lan Huang[1,4], Andre J Ouellette[5], Albert Pol[3,6], Michael A Welte[2], Steven P Gross[1]\***

[1]Department of Developmental and Cell Biology, University of California Irvine, Irvine, United States; [2]Department of Biology, University of Rochester, Rochester, United States; [3]Equip de Proliferació i Senyalització Cel.lular, Institut d'Investigacions Biomèdiques August Pi i Sunyer (IDIBAPS), Barcelona, Spain; [4]Department of Physiology and Biophysics, University of California Irvine, Irvine, United States; [5]Department of Pathology and Laboratory Medicine, Keck School of Medicine, University of Southern California, Los Angeles, United States; [6]Institució Catalana de Recerca i Estudis Avançats, Barcelona, Spain

**Abstract** We previously discovered histones bound to cytosolic lipid droplets (LDs); here we show that this forms a cellular antibacterial defense system. Sequestered on droplets under normal conditions, in the presence of bacterial lipopolysaccharide (LPS) or lipoteichoic acid (LTA), histones are released from the droplets and kill bacteria efficiently in vitro. Droplet-bound histones also function in vivo: when injected into *Drosophila* embryos lacking droplet-bound histones, bacteria grow rapidly. In contrast, bacteria injected into embryos with droplet-bound histones die. Embryos with droplet-bound histones displayed more than a fourfold survival advantage when challenged with four different bacterial species. Our data suggests that this intracellular antibacterial defense system may function in adult flies, and also potentially in mice.

**\*For correspondence:**
sgross@uci.edu

[†]These authors contributed equally to this work

**Competing interests:** The authors have declared that no competing interests exist

**Reviewing editor**: Roberto Kolter, Harvard Medical School, United States

## Introduction

Histones are fundamental components of eukaryotic chromatin, and are therefore abundant in essentially all animal cells, for example, in humans, there are tens of millions of histone molecules per cell. While not generally appreciated, histones and histone fragments are surprisingly bactericidal in in vitro assays (*Hirsch, 1958*). Thus, in principle histones could provide animals with a potent supply of microbicides for protection against bacterial invasion. In some cases, histones are released extracellularly, and then contribute to innate immunity against bacteria. For example, in the skin mucosa of catfish, secreted histone H2A contributes critically to the organismal defense against bacteria (*Cho et al., 2002*), and histones H3 and H4 are prominent antimicrobial agents in sebaceous gland secretions (*Lee et al., 2009*).

Many bacterial pathogens invade cells and replicate intracellularly. This infectious strategy allows evasion of many host innate and adaptive immune mechanisms, including antibody-based defenses in vertebrates. In principle, then, histones normally present in cells could confer protection against microbes that exploit intracellular niches. However, until now this has seemed unlikely for two reasons. First, histones typically are located in the nucleus, excluding contact with cytosolic bacteria. Second, histones are generally believed to be predominantly bound to DNA, with only minuscule amounts of excess histones. In fact, because unconfined histones cause genomic instability, hypersensitivity to DNA-damaging agents, and lethality (*Saffarzadeh et al., 2012*), cells minimize excess free histones by active mechanisms (*Singh et al., 2009a, 2009b*). Given these constraints, it is not surprising that histone-mediated intracellular antibacterial responses have not been investigated extensively.

**eLife digest** Histones are proteins found in large numbers in most animal cells, where their primary job is to help DNA strands fold into compact and robust structures inside the nucleus. In vitro, histones are very effective at killing bacteria, and there is some evidence that histones secreted from cells provide protection against bacteria living outside cells. However, many types of bacteria are able to enter cells, where they can avoid the immune system and go on to replicate.

In principle histones could protect cells against such bacteria from the inside, but for many years this was thought to be unlikely because most histones are bound to DNA strands in the cell nucleus, whereas the bacteria replicate in the cytosol. Moreover, free histones can be extremely damaging to cells, so most species have developed mechanisms to detect and degrade free histones in the cytosol.

Recently, however, it was discovered that histones can bind to lipid droplets—organelles in the cytosol that are primarily used to store energy—in various animal cells and tissues. Now, Anand et al. have demonstrated that histones bound to lipid droplets can protect cells against bacteria without causing any of the harm normally associated with the presence of free histones. In in vitro experiments with lipid droplets purified from *Drosophila* embryos, they showed that histones bound to lipid droplets could be released to kill bacteria. The histones were released by lipopolysaccharide or lipoteichoic acid produced by the bacteria.

The effect was also observed in vivo: using four different bacterial species, Anand et al. injected similar numbers of bacteria into *Drosophila* embryos that contained histones bound to lipid droplets, and also into embryos that had been genetically modified so that they did not contain such droplet-bound histones. While most of the normal embryos survived, the vast majority of the embryos without droplet-bound histones died. Similar results were also found in experiments on adult flies, along with evidence which suggests that histones might also provide defenses against bacteria in mice.

Lipid droplets (LDs) are ubiquitous fat storage organelles that store triglycerides and sterols for energy production and as biosynthetic precursors. In addition to their fundamental role in lipid homeostasis, LDs have recently been proposed to act as protein sequestration sites (*Cermelli et al., 2006*; *Welte, 2007*). Histones, in particular, have been detected on LDs in a number of animal cells and tissues, including *C. elegans*, fly embryos, moth fat bodies, and mammalian leukocytes, insulin producing β-cells, and muscles (*Cermelli et al., 2006*; *Wan et al., 2007*; *Yang et al., 2010*; *Zhang et al., 2011*; *Larsson et al., 2012*; *Zhang et al., 2012*). Might these cytosolic extra-nuclear histone deposits allow cells to exploit the antimicrobial properties of histones without incurring the risks typically caused by excess histone accumulation?

We address this question using early *Drosophila* embryos. In these embryos, LDs sequester large amounts of histones (*Cermelli et al., 2006*), and new work has provided mutants lacking droplet-bound histones (*Li et al., 2012*). Here, we report that while histones are usually sequestered on droplets, they are released in response to the presence of bacterial cell wall components. These histones kill Gram-positive as well as Gram-negative bacteria, both in vitro and in vivo. Embryos injected with bacteria have considerably higher survival success when they have droplet-bound histones, establishing that histones on LDs contribute to innate immunity in embryos. A similar survival advantage was observed in adult flies. Simulation of bacterial infection in mice induces histone accumulation on LDs in the liver, a key organ for fighting infection; this observation suggests that lipid droplet histones may represent an ancient host defense strategy.

## Results

### LDs have antimicrobial activity

Our earlier study (*Cermelli et al., 2006*) established the presence of histones (usually believed to be nuclear) in cytosolic lipid droplets purified from *Drosophila* embryos. Based on our discovery of immunity-related mRNAs on the LDs (unpublished), we considered whether histones might play a role in the flies' immune response. When we found that there was a precedent for histone antibacterial activity (*Hirsch, 1958*), we hypothesized that the histones bound to the cytosolic LDs might act as an

antibacterial system. To test whether LDs could indeed inhibit bacterial growth, we performed a traditional plate assay (*Figure 1A*). Dilute suspensions of bacteria were grown in the presence or absence of potential antimicrobial agents (LDs equivalent to ~500 µg total proteins and controls), and colony forming units (CFU) on an agar plate were counted after 24 hr of incubation with the agents. For both Gram-negative (*Escherichia coli*) and Gram-positive (*Staphylococcus epidermidis*) bacteria, LDs decreased the CFU dramatically (*Figure 1A*, compare buffer vs LD), indicating that the droplets have an antimicrobial property. These effects were highly reproducible in multiple trials (*Figure 1B*). Complementary disc-diffusion assays confirmed the microbicidal effects of the droplets (*Figure 1C,D*).

What is the molecular nature of this killing activity? LDs are complex organelles with both lipids and proteins. Thus, one potential source of antibacterial activity might be fatty acids released due to breakdown of the abundant triglycerides; because of their detergent-like properties, they might destroy bacterial membranes and thus impair bacterial viability. We therefore treated purified LDs with alkaline washes. This treatment removes electrostatically bound droplet proteins, but not lipids and proteins attached via hydrophobic interactions (*Brasaemle et al., 2004*). Indeed, as previously reported (*Cermelli et al., 2006*), these washes greatly reduced the levels of the electrostatically bound histone H2B, but not of the hydrophobically bound LSD-2. These treated droplets no longer had antibacterial activity (*Figure 1C*, LD-CaCO$_3$), suggesting that the antimicrobial activity requires electrostatically bound proteins.

Our previous study identified hundreds of lipid droplet proteins (*Cermelli et al., 2006*), many present in low enough copy number to be detectable only by silver stain. In principle, any of them might be responsible for the antibacterial activity, but it seems likely that the active agents would be present in high copy number. We thus performed a traditional gel overlay assay (*Figure 1E*), visualizing these high-copy number proteins by Coomassie stain. Droplet proteins were separated by Acid Urea (AU) gel electrophoresis, and the gel was overlain with nutrient agar seeded with bacteria. As in the disc assay, antibacterial proteins diffuse into the agar and locally inhibit bacterial growth, as shown by the positive control mouse α-defensin cryptdin-4 (*Figure 1E*, gel overlay violet arrow). The lipid-droplet lane contained significant antibacterial activity in only one location (*Figure 1E*, gel overlay red arrow). MS analyses of proteins eluted from this location identified histones H2A and H2B as the predominant proteins present (spectra not shown). Therefore, the bactericidal activity of LDs is most likely due to histones, consistent with their known antimicrobial activities in vitro (*Hirsch, 1958*). Importantly, pretreatment of purified LDs with anti-histone antibodies abolished or markedly reduced the droplet bactericidal activity (*Figure 1A*, 1B-LD + Anti-histones). Thus, we conclude that the majority of the in vitro antibacterial activity of LDs from early *Drosophila* embryos is due to histones.

We performed two tests to determine if these cytosolic droplet-bound histones are different from nuclear histones. First, using mass spectrometry, we compared post-translational modifications on droplet-bound and nuclear histones and found no major distinctions, suggesting that unique post translation modifications are not responsible for association of histones to LDs. Since our analysis was mostly qualitative, we cannot rule out the possibility that differences might arise from quantitative changes in post-translation modifications. We did identify several acetylation sites in histones H2A (serine 1 and lysines 5 and 8) and histone H2B (lysines 7, 11, 14 and 17). Some of these acetylation sites were previously found in shrimp histones (*Ouvry-Patat and Schey, 2007*). The histones previously examined for antimicrobial activity are reported to be un-acetylated (*Kim et al., 2000*), though there was no indication of whether acetylation affected antimicrobial efficacy. Second, we compared the antibacterial potencies of droplet-derived histones with commercial calf thymus histones isolated from a calf thymus nuclear fraction in both bactericidal (killing bacteria outright) and bacteriostatic (inhibiting the growth or reproduction of bacteria) activity assays. Histones extracted from droplets using AU gel electrophoresis were combined with a suspension of *E. coli* ML35 for 1 hr, and bacterial cell survival was determined by measuring CFU. This assay showed that histones are bactericidal and that droplet-derived histones and commercial pan-histones purified from calf thymus do not differ significantly in antibacterial efficacy (*Figure 1F*).

## Enhanced bacterial growth in embryos lacking histone deposits on LDs

Although these studies indicate that histones present on embryonic LDs kill bacteria in vitro, it remained unclear whether histones make a meaningful contribution to the overall antibacterial defense in the embryos. In vivo, droplet-bound histones may have different properties, due to the presence of binding partners or the physiological state of the bacteria, their effective concentration might not be

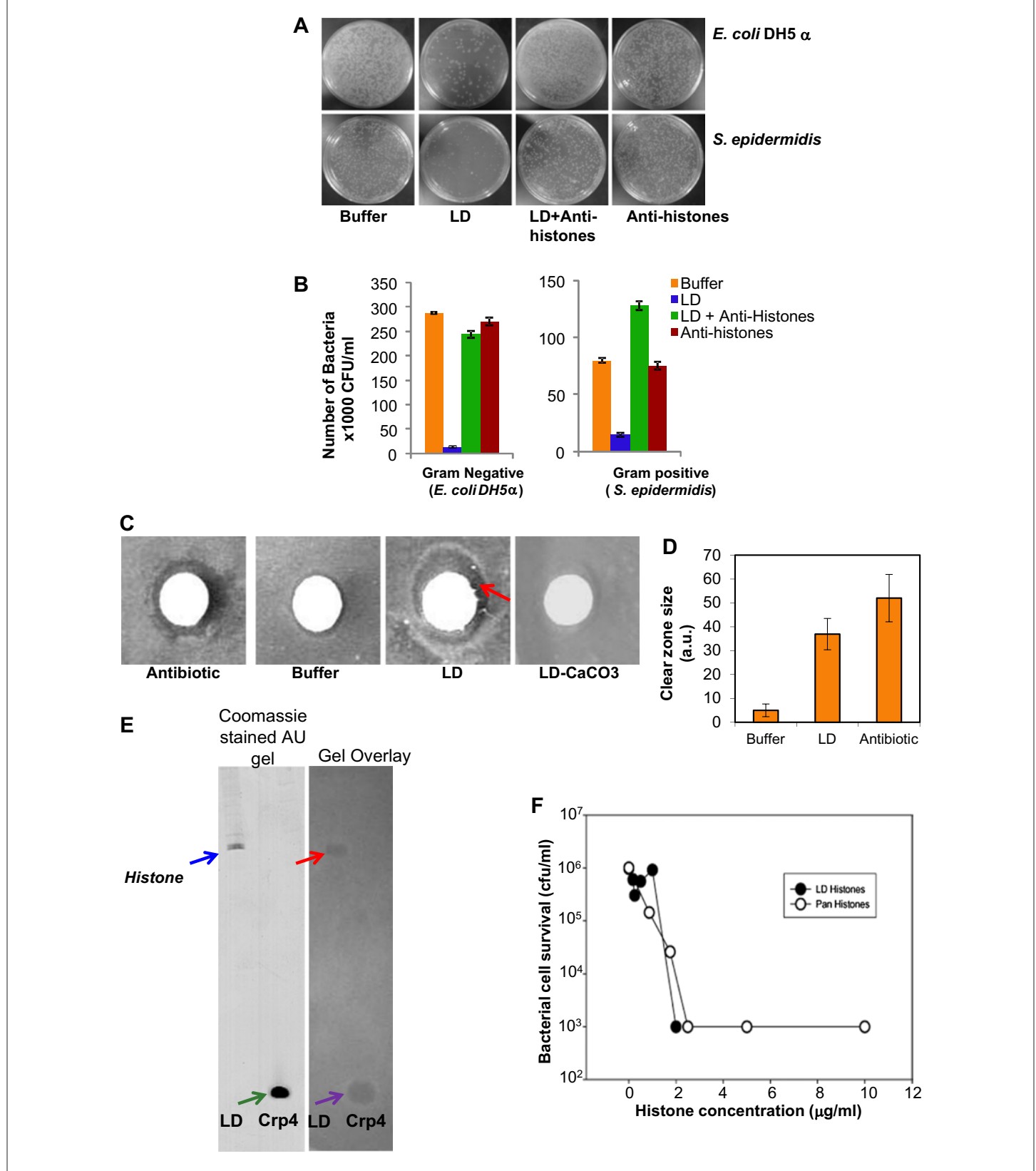

**Figure 1**. LDs kill bacteria via droplet bound histones. (**A**). Representative plates in a colony forming assay, showing growth of Gram-negative (*Escherichia coli DH5α*, top) and Gram-positive (*Staphylococcus epidermidis*, bottom) bacteria, where a known amount of bacteria were incubated at

*Figure 1. Continued on next page*

*Figure 1. Continued*

37°C either in buffer alone, or with LDs pre-treated with or without anti-histone antibodies. In buffer (left, 'Buffer'), many colonies (white spots) were observed, but in the presence of LDs (LD), the observed number of colonies was greatly decreased, demonstrating an antibacterial effect of the LDs. Pre-treatment of the droplets with anti-histone antibodies abolished this effect (LD + Anti-histones). (**B**). Quantification of colony forming assay in **A**. Each bar represents the mean number of observed colonies, in three independent trials, presented with the standard error. (**C**). Disc diffusion assay over a lawn of *E. coli* DH5α. A potential antibacterial agent is placed on a small sterile piece of filter paper (white circle); a cleared area (darker region) indicates antibacterial activity. Positive control: the antibiotic kanamycin (Antibiotic); growth inhibition region indicated by the red arrow. Negative control: buffer (Buffer). LDs isolated in the presence (LD-CaCO₃) or absence (LD) of alkaline carbonate were spotted on sterile discs; bacterial inhibition was observed in the untreated droplets (LD) but not in the carbonate-treated droplets (LD-CaCO₃). The filter papers are 7 mm in diameter. (**D**). Quantification of the size of the clear zone in the disc diffusion assay from **C**. Fifteen independent disc diffusion assays were performed with purified LDs from *Drosophila* embryos, the antibiotic kanamycin, or buffer. Antimicrobial activity of compounds was quantified as the diameter of the clear zones surrounding the filter papers after subtraction of filter papers diameter (1 A.U. = 0.1 mm). (**E**). Use of a gel-overlay assay to determine the identity of the anti-bacterial protein(s) on the LDs. Proteins extracted from LDs (LD, left lane) were run in duplicate on an AU-gel; murine cryptdin 4 (Crp 4, right lane) served as positive control. After electrophoresis, the gel was split. One half (left) was stained by Coomassie Blue; the histone bands are indicated by a blue arrow and the crp 4 control is indicated by the green arrow. The other half (right) was used in a gel overlay assay (see 'Materials and methods') to reveal regions of the gel able to inhibit bacterial growth (inhibition by LD is indicated by the red arrow and that by crp 4 control is indicated by the violet arrow). Inhibition of bacterial growth due to proteins on the LDs was only observed in a single region, corresponding to the histones (red arrow). Consistent with this, mass spectrometry of proteins cut from the Coomassie gel corresponding to the killing region identified predominantly histones H2A and H2B (see 'LDs have antimicrobial activity'). (**F**). *E. coli* ML35 cultures were transiently (~1 hr) exposed to commercial calf thymus pan-histone proteins (Sigma) or gel-extracted LD-histones (from the gel-overlay assay). Both preparations show similar potency for bacterial killing.

high enough to kill, or, relative to other antibacterial mechanisms, the contribution of histones might be negligible.

To test the significance of the histones on LDs in vivo, we took advantage of the recent identification of the putative histone receptor on droplets, jabba (CG42351), a novel 42 kDa protein (*Li et al., 2012*). *Jabba* is present on LDs in wild-type flies and is required for histone localization on droplets; in *Jabba* mutants LDs are present, but histones are absent from the LDs (*Figure 2*). We used two *Jabba* alleles, derived independently. *Jabba^{zl01}* is due to imprecise excision of a P element inserted near the *Jabba* promoter, and *Jabba^{f07560}* is due to the insertion of a PBac element in the middle of the *Jabba* coding region. The two alleles were generated from entirely different genetic backgrounds, ruling out genetic background effects. In the wild type, fusions between GFP and the histone H2Av are present both in nuclei and in cytoplasmic rings (*Figure 2A*), a characteristic appearance of droplet-targeted proteins (*Cermelli et al., 2006*), but cytoplasmic rings are undetectable in *Jabba* embryos. Second, when living embryos are centrifuged, LDs separate from the rest of the embryonic content and form a distinct layer (*Cermelli et al., 2006*). Immunostaining of such centrifuged embryos reveals abundant histone signal in the droplet layer in the wild type, but not in *Jabba* embryos (*Figure 2B,C*). Third, Western blotting reveals high levels of histones on droplets purified from wild-type embryos, but not on droplets purified from *Jabba* mutants (*Figure 2D,E*). Such purified droplets also differed in their antimicrobial activity in vitro: droplets from wild-type embryos resulted in a 10-fold decrease in bacterial growth relative to buffer alone, but droplets from two different independently isolated *Jabba* mutant strains displayed essentially no killing activity (*Figure 2G*), as expected if the killing activity of wild-type droplets is indeed due to histones.

In *Jabba* mutant embryos, the overall levels of histones are much reduced relative to the wild type (*Figure 2F*), presumably because the histones not sequestered on the droplets are degraded. At least in other systems, unconfined histones are rapidly eliminated via proteolysis (*Singh et al., 2009a, 2009b*), a protective mechanism against the detrimental effects of free histones.

*Jabba* mutants are viable and fertile and develop into apparently healthy adults (though likely with compromised immune systems, see *Figure 4*). In particular, embryos hatch at wild-type rates. These mutants therefore make it possible to ask if the extra-nuclear pool of histones affects the outcome of bacterial infections in vivo. We microinjected (*Figure 3A*) very early wild-type or *Jabba* embryos (less than 1-hr old) with a GFP-expressing *E. coli* strain; living bacteria are easily identified by GFP fluorescence; fluorescence fades after bacterial cell death (*Lowder et al., 2000*). Monitoring overall GFP fluorescence provided a measure of live bacteria in the infected embryos. The injection protocol per se does not apparently harm either genotype, since buffer-only injection resulted in high and similar hatching success (*Figure 3B*).

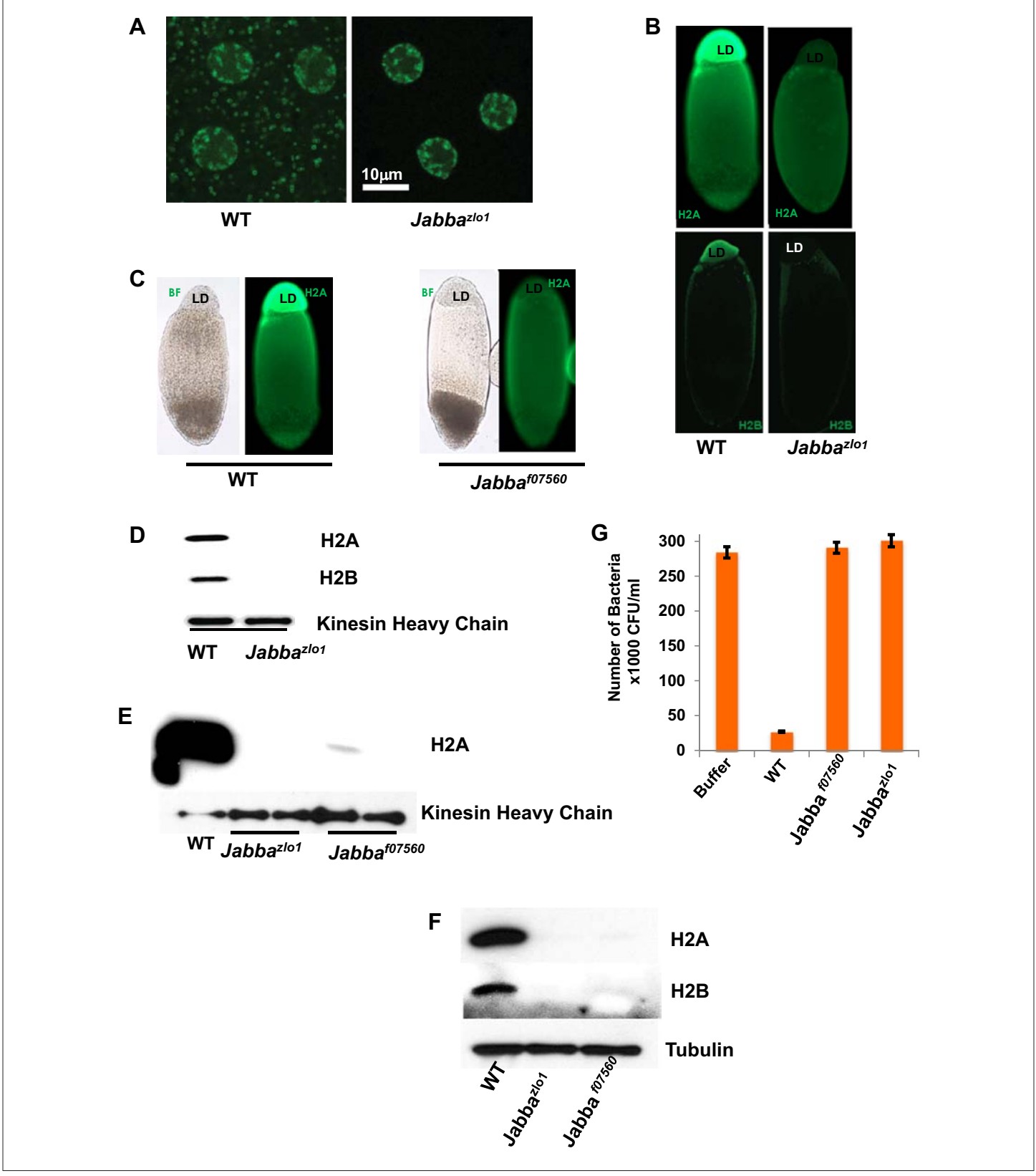

**Figure 2**. Presence of extranuclear histones depends on the *Jabba* protein. (**A**) Histone H2Av GFP is not detectable in cytoplasmic puncta of *Jabba^zl01^* embryos. Both genotypes show strong signal in nuclei. (**B**). By immunostaining, endogenous H2A and H2B are absent from the lipid droplet layer (LD) of centrifuged *Jabba^zl01^* embryos. (**C**). Histone H2A is absent from the lipid droplet layer in centrifuged *Jabba^f07560^* embryos. BF is the bright field image and
*Figure 2. Continued on next page*

*Figure 2. Continued*

LD is the lipid droplet layer. (**D**). Equal amounts of proteins from purified LDs were compared by Western analysis. Droplets from *Jabba^zl01* embryos lack histones H2A and H2B. The droplet-bound Khc protein serves as loading control. (**E**). When compared side by side, similar reductions in droplet-bound histones were found for both the independently isolated Jabba alleles *Jabba^f07560* and *Jabba^zl01*. (**F**). Western blot of equal numbers of unfertilized wild-type and *Jabba* mutant embryos. Overall levels of histone H2A and H2B are significantly reduced in the *Jabba* mutants. (**G**). LDs purified from embryos of two independently isolated *Jabba* mutants revealed no bacterial killing activity in antibacterial plate assays, with bacterial growth comparable to buffer alone, in contrast to droplets purified from wild-type embryos which dramatically decreased bacterial growth.

Under the conditions employed, bacterial numbers in wild-type embryos decreased with time (*Figure 3C*). 2 hr after injection, numerous individual bacteria were obvious in the embryos, and we estimate on the order of 84 bacteria per embryo (see 'Materials and methods'). By 24 hr, the number of visible bacteria had decreased substantially, and we estimate on the order of 33 bacteria per embryo. By 48 hr, there typically were either no surviving bacteria or only a few detectable bacterial cells, and on average we estimate 2–6 bacteria were present. Thus, some innate immune mediator(s) limits bacterial cell viability in wild-type embryos in this experimental system.

This limit on bacterial viability was lost in the *Jabba* embryos (*Figure 3C*). The appearance of multiple individual bacteria 2 hr post-infection was similar to the wild type, and we estimate approximately 79 bacteria per embryo. However, bacterial cell numbers increased dramatically by 24 hr, with 840 bacteria on average. By 48 hr, we estimate that thousands of bacteria were present in the *Jabba* embryos (*Figure 3C*, compare WT to *Jabba^f07560* at 24 and 48 hr; also see *Figure 3—figure supplements 1–3*). These results indicate that loss of histones on droplets due to the absence of functional *Jabba* correlates with susceptibility to massive bacterial overgrowth. Because bacterial numbers are controlled, and bacteria are ultimately eliminated in wild-type embryos, it suggests that droplet-associated histones contribute to immunity against the introduced bacteria, consistent with their in vitro bactericidal capability.

## Droplet-bound histones enhance embryo survival in response to a bacterial challenge

It seemed likely that the immunity observed in this experimental bacterial infection would have consequences for the embryo. To test this, we challenged wild-type and *Jabba* mutant embryos with different bacterial species (*Figure 3A*) and assessed the effects of genotype on survival of the embryos (*Figure 3D*). We first injected embryos with two laboratory strains of bacteria: Gram-negative *E. coli* DH5α and Gram-positive *S. epidermidis*. In each case, experiments were done in parallel, with 50–100 bacteria being injected into multiple embryos of each genotype. Embryos that hatched into larvae were scored as surviving. Injection of bacteria delayed wild-type embryonic development, but caused only a minimal increase in lethality, relative to buffer-only injected embryos (normalized survival more than 80%; see *Figure 3D*, *S. epidermidis* and *E. coli* DH5α, orange bars). In contrast, for the two independently derived *Jabba* mutants (both lacking droplet-bound histones; *Figure 2D*), the same treatment resulted in high lethality, with normalized survival of less than 20% (*Figure 3D*). Thus, *Jabba* embryos exhibited at least a fourfold decrease in survival when injected with either species of bacteria. Such a difference in survival would provide a huge survival advantage in nature.

*E. coli* and *S. epidermidis* do not typically grow intracellularly, but bacterial pathogens that grow intracellularly are not well characterized in flies. We therefore took advantage of two species of bacteria with well-characterized intracellular mechanisms of infection, *Bacillus subtilis* engineered to express *Listeria*'s hemolysin-A protein (*Bielecki et al., 1990*), and thus able to enter cells and reproduce in the cytosol, and also *Listeria monocytogenes* whose infectious life cycle typically involves growth in the cytosol of mammalian cells (*Tilney and Portnoy, 1989*). At moderate injection dosages of 50–100 bacteria per embryo, there was again good survival for wild-type embryos, but not for the *Jabba* embryos (*Figure 3D*, *B. subtilis* and *L. monocytogenes*). In conclusion, a marked survival difference between the wild-type (more survival) and *Jabba* mutant embryos was observed when infected with bacteria, for all of the bacterial species tested.

## Adult flies with droplet-bound histones show enhanced survival in response to a bacterial challenge

Could *Jabba* function in adults? Consistent with a possible role in facilitating stable localization of histones to the cytoplasm, *Jabba* is expressed in a variety of adult tissues and in both sexes, with

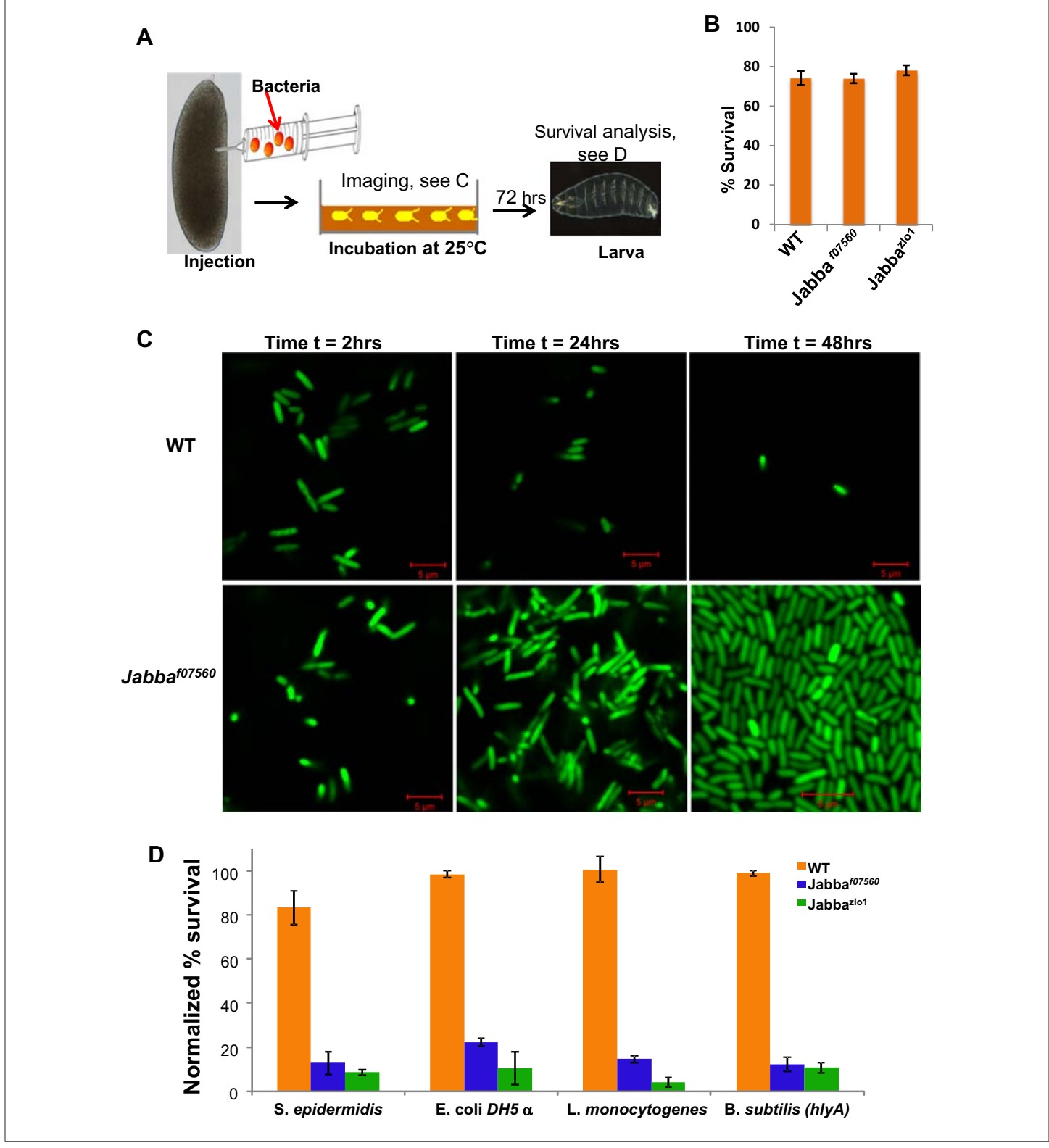

**Figure 3**. LD bound histones can kill bacteria in vivo. (**A**) Schematic representation of embryo microinjection. Early embryos collected within half an hour of laying were injected with a bacterial suspension, as detailed in 'Materials and methods'. (**B**). Wild-type and *Jabba* mutant embryos show similar survival when injected with buffer alone. Wild-type and *Jabba* mutants (*Jabba^f07560^*, *Jabba^zl01^*) embryos were injected with microinjection buffer (no bacteria) and the percentage survival was scored 72 hr post injection. (**C**) Bacteria grow only in embryos lacking droplet-bound histones. Approximately
*Figure 3. Continued on next page*

*Figure 3. Continued*

equal numbers of GFP labeled bacteria (*E. coli* strain YD133) were injected into wild-type and *Jabba* mutant embryos (*Jabba^f07560*) and the growth of bacteria inside embryos was monitored at various times post injection. (**D**). *Drosophila* embryos lacking droplet-bound histones have reduced survival due to bacterial infection. Approximately equal numbers of bacteria were injected into wild-type and *Jabba* mutant (*Jabba^zl0* and *Jabba^f07560*) embryos and embryo survival after 72 hr was normalized to the buffer-only injected embryos (in **B**). The bacterial strains used were *Staphylococcus epidermidis* (Gram-positive); *E. coli* DH 5α (Gram-negative); *Listeria monocytogenes* (Gram-positive and intracellular); and *Bacillus subtilis (hlyA)*, modified *Bacillus subtilis* expressing *listeria* hemolysin-A protein (Gram-positive and intracelluar).

The following figure supplements are available for figure 3.

**Figure supplement 1**. 2 hr: Additional images of wild-type and Jabba mutant embryos with fluorescent bacteria, 2 hr after bacterial injection.

**Figure supplement 2**. 24 hr: Additional images of wild-type and Jabba mutant embryos with fluorescent bacteria, 24 hr after bacterial injection.

**Figure supplement 3**. 48 hr: Additional images of wild-type and Jabba mutant embryos with fluorescent bacteria, 48 hr after bacterial injection.

especially high expression levels in fat body and ovaries, according to microarray and RNA-seq data available on FlyBase (*Chintapalli et al., 2007*; *McQuilton et al., 2012*).

To test whether this *Jabba* protein present in the adults might contribute to a similar LD-histone system, we used a traditional bacterial challenge assay, where bacteria were introduced into adult flies by pricking the flies under the wing with a metallic needle dipped in either sterile buffer or a concentrated bacterial suspension. Pricking either the wild type (black curve, *Figure 4A,B*) or *Jabba*-mutant adults (red curves, *Figure 4A,B*) with the buffer-dipped needle resulted in low long-term mortality, with roughly a 20–30% mortality at 4 days. At the dose of *Listeria* used, mortality of the pricked wild-type adults was approximately the same as the buffer-pricked adults (the purple mortality curve in *Figure 4A* is within experimental error of the black curve). However, for the *Jabba*-mutant adults, pricking with a bacterial-dipped needle was quite lethal (*Figure 4A,B*, brown curve), with less than 5% survival at 4 days—a 14-fold difference from wild-type survival.

How likely is it that the underpinnings of the survival difference reflect the same mechanism? We first examined relative bacterial load via a plate assay using cytoplasmic extract from the buffer or bacterial-pricked adults to seed the plate. From wild-type or *Jabba*-mutant adult buffer-pricked cytoplasm, typically less than three colonies were observed. In contrast, for the bacterial-pricked adults, initial counts were typically on the order of 400 CFUs (*Figure 4C*), and by day 3 went down significantly for the wild type (50) but less so for the surviving *Jabba*-mutant flies (320). Presumably, the *Jabba*-mutant flies that died (not assayed) had even higher bacterial counts. While a complete investigation of the kinetics of bacterial load is beyond the scope of this paper, as in the embryos, these results suggest that the observed lethality correlates with differences in bacterial load.

Finally, we looked for the presence of histones in the adult cytoplasm. First, cytoplasmic lysates were made from 1- to 2-day-old adult wild-type or *Jabba*-mutant flies as detailed in 'Materials and methods', and then equal amounts of the lysates from both classes of adults were blotted to detect histone H2B (*Figure 4D*). Compared to the wild-type, the amount of H2B detected was lower in the *Jabba*-mutant background (threefold), consistent with the embryo data. In conclusion, while more work remains to understand the role of *Jabba* and histones in adult immunity, our initial data is consistent with the hypothesis that the embryonic system described above may function in adult flies as well.

## How histones reach bacteria: selective release

Since excess free histones are deleterious for the cell overall (*Gunjan and Verreault, 2003*), droplet-bound histones are likely relatively immobilized: we expected them to be sequestered on droplets and not free to diffuse. Indeed, when purified droplets are incubated in excess buffer, there is no detectable loss of histones from the droplets, or appearance of histones in the buffer (*Figure 5A,B*, UB control). Thus, the histones indeed appear to be stably bound to LDs. This might limit their ability to reach the bacteria since the diffusion constant of a 0.5-μm droplet is expected to be much lower than the diffusion constant of a free histone.

These observations are seemingly contradictory: histones are stably bound to droplets, yet they can kill bacteria well. We thus hypothesized that the bacteria may induce release of the histones from the

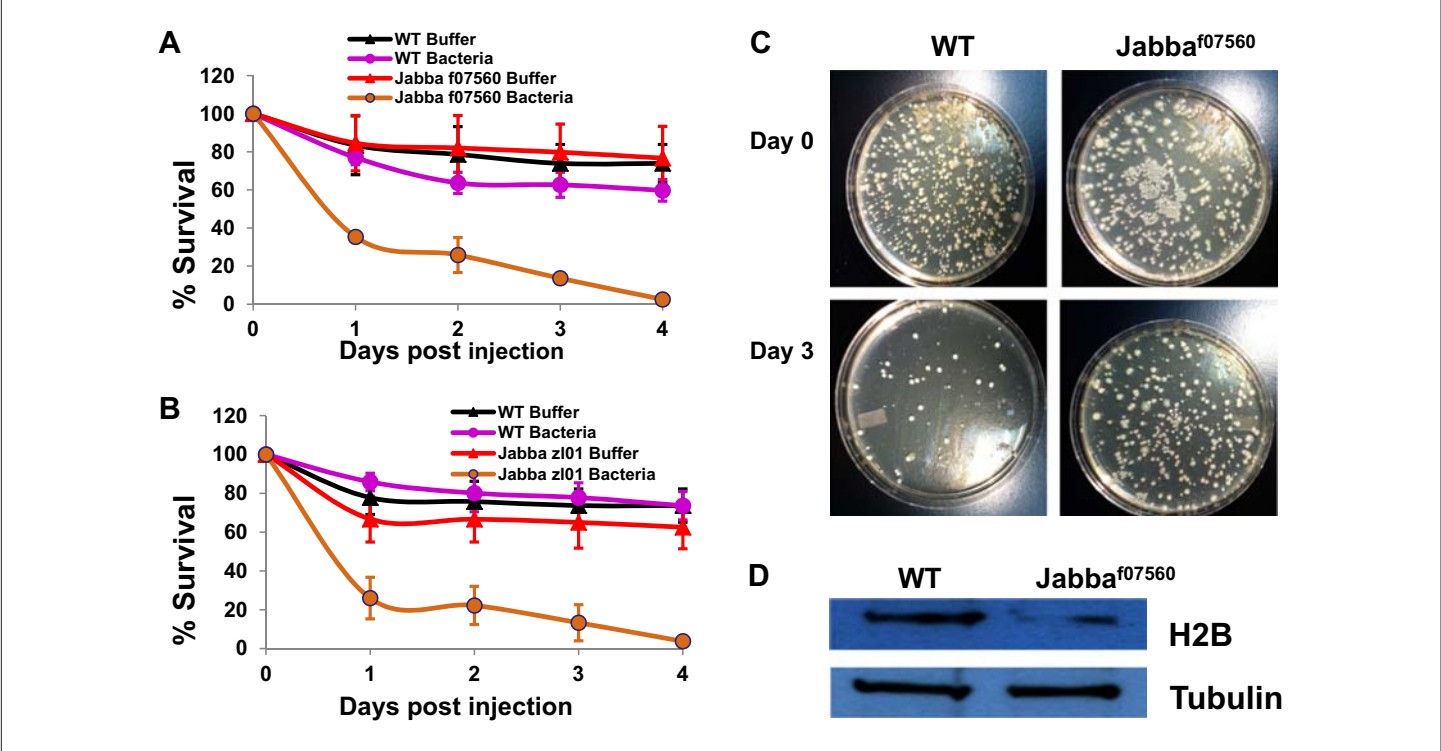

**Figure 4**. The Jabba protein contributes to improved survival for adult flies. (**A** and **B**) Adult *Drosophila* lacking *Jabba* (**A**: *Jabba^f07560^*; **B**: *Jabba^zl01^*) have reduced survival when challenged by bacteria. Wild-type and *Jabba* mutant (*Jabba^zl0^ and Jabba^f07560^*) adult flies were infected with *Listeria monocytogenes* as detailed in 'Materials and methods', and fly survival was monitored over the course of 4 days. (**C**). Representative plates in a colony forming assay, showing bacterial colonies on agar plates streaked with cytosolic extract from bacteria infected adult flies. (**D**) Western blots of histone H2B from equal amounts of cytosolic extracts from wild type and *Jabba* mutant adult flies, showing that overall levels of H2B were significantly reduced in the *Jabba* mutants.

droplets. Pathogen-Associated Molecular Patterns, components of the bacterial envelope, would be particularly well positioned to induce such a release, as they are present on the surface of bacteria and thus accessible. Indeed, organisms often detect bacterial infections due to the presence of LPS (*Heumann and Roger, 2002*) or LTA (*Wergeland et al., 1989*), major pro-inflammatory constituents of Gram-negative and Gram-positive bacterial cell envelopes, respectively.

We therefore incubated purified LDs in the presence or absence of LPS or LTA. Histones were detected in the buffer (UB, *Figure 5A,B*) only when LPS or LTA were included, and histone amounts increased with increasing levels of the cell envelope components (*Figure 5A,B*); concomitantly, histones attached to the LDs decreased (LD, *Figure 5A,B*). Thus, LPS and LTA induce release of histones from the droplets in a dose-dependent manner.

## Potential evolutionary conservation: infection increases droplet-bound histone H1 in mice

Histones on LDs are not restricted to *Drosophila*. In particular, specific histones have been identified on LDs purified from a number of mammalian cell lines and tissues (*Smolenski et al., 2007*; *Wan et al., 2007*; *Zhang et al., 2011*; *Larsson et al., 2012*). Thus, this defense system may be widely conserved.

As a preliminary test, we looked at droplets in the liver, as this organ removes pathogens and microbial products from the blood, and plays a key role in the body's immune response (*Mackay, 2002*). LDs were purified from murine liver using a previously established protocol (*Turro et al., 2006*); the hepatocyte lipid-droplet resident protein (*Turro et al., 2006*) ALDI was enriched 10³-fold (*Figure 6A*), confirming the success of the fractionation. By Western blotting, we detected histone H1 in the droplet fraction, using three different specific antibodies generated in different species (*Figure 6B*). This

**Figure 5**. Bacterial cell wall components release droplet bounds histones in a dose dependent manner. (**A**). Increasing concentrations of lipopolysaccharide (LPS) in the buffer releases droplet bound histones from purified LDs. (**B**). Lipoteichoic acid (LTA) causes the dose dependent release of histones from purified droplets. LDs were purified from wild-type *Drosophila* embryos, re-suspended in buffer, and incubated for 2 hr at room temperature with different concentrations of LPS or LTA. LDs (LD) were then separated from the under-layer buffer (UB) and both were processed for SDS-PAGE. Western Blot analysis was carried out with H2B histone antibodies.

does not represent contamination with nuclei or chromatin since histones H2A, H2B and H3 were not detectable (*Figure 6A*). Normalized for total protein, the H1 levels on LDs were comparable to the levels of H1 histone in a purified nuclear fraction (*Figure 6B*—n), and were clearly much higher than in whole-liver homogenates (*Figure 6B*—h). Like other histones, histone H1 also has antibacterial activity in vitro (*Parseghian and Luhrs, 2006*).

Intriguingly, the levels of histone H1 on LDs increased under conditions that mimicked a systemic infection. Mice were intraperitoneally injected with LPS (20 ng), and assayed 16 hr later. LPS injection resulted in the expected hepatic injury (*Senga et al., 2008*), as assessed by the increased presence of serum transaminases (AST and ALT) and cytokines (IL-6 and TNFα; *Figure 6C*). Relative to untreated controls, H1 levels in the lipid-droplet fraction were clearly increased in the LPS challenged animals (average ratio of H1 in LPS injected to H1 in uninjected was $1.6 \pm 0.3$ (mean $\pm$ SEM), significantly different with a $p=0.025$ by Student t-test; note that the amount of the lipid droplet marker protein ALDI went down slightly in the LPS-injected samples, and if we use this as a standard, and normalize the detected ratio, it becomes 1.86 instead of 1.6). Like the *Drosophila* histones, H1 was efficiently released from the LDs by LPS (*Figure 6D*), consistent with the hypothesis that the accumulated H1 on the hepatic LDs can be released by the presence of cytosolic bacteria, and thus might contribute to an antibacterial response.

These two observations—that systemic LPS increases histones on murine droplets, but that direct treatment of purified droplets with LPS releases the histones—might appear contradictory, but we believe they are not. In the first case, the systemic LPS is known to activate numerous defense pathways which could then result in increased loading of droplets with histones. Further, lacking cytosolic bacteria, the systemic circulating LPS is unlikely to be present inside the cells at high enough dosage to release the histones from the droplets. In the second case, purified droplets with histones are exposed directly to LPS, mimicking the presence of bacteria inside of cells. In this latter case, there are no signaling effects, because in the purified system no signaling apparatus is present. Thus, this second assay looks only at the direct effect of high levels of LPS interacting with histones, causing the histones to release from the droplets.

## Discussion

Here, we suggest that LDs contribute to host defense by sequestration and regulated release of histones. Analogous to mammalian mothers depositing antibodies across the placenta into the embryo as it develops, we suspect that maternal deposition of the histone laden lipid droplets into the embryo may prevent transmission of any bacterial infection from mother to egg, and may also protect the embryo from subsequent bacterial infections, resulting, for example, from physical damage to the chorion.

The histones are stably bound to *Drosophila* and murine LDs, but bacterial cell envelope components promote their release. As tested in the embryos, and in a limited way in adult flies, the system appears efficient, resulting in a more than fourfold improvement in survival for challenges by multiple types of Gram-positive and Gram-negative bacteria, including the known intracellular pathogen *Listeria*. Histones and histone fragments had previously been reported to play a role in extracellular antibacterial defense (*Fernandes et al., 2004*; *von Kockritz-Blickwede and Nizet, 2009*). We

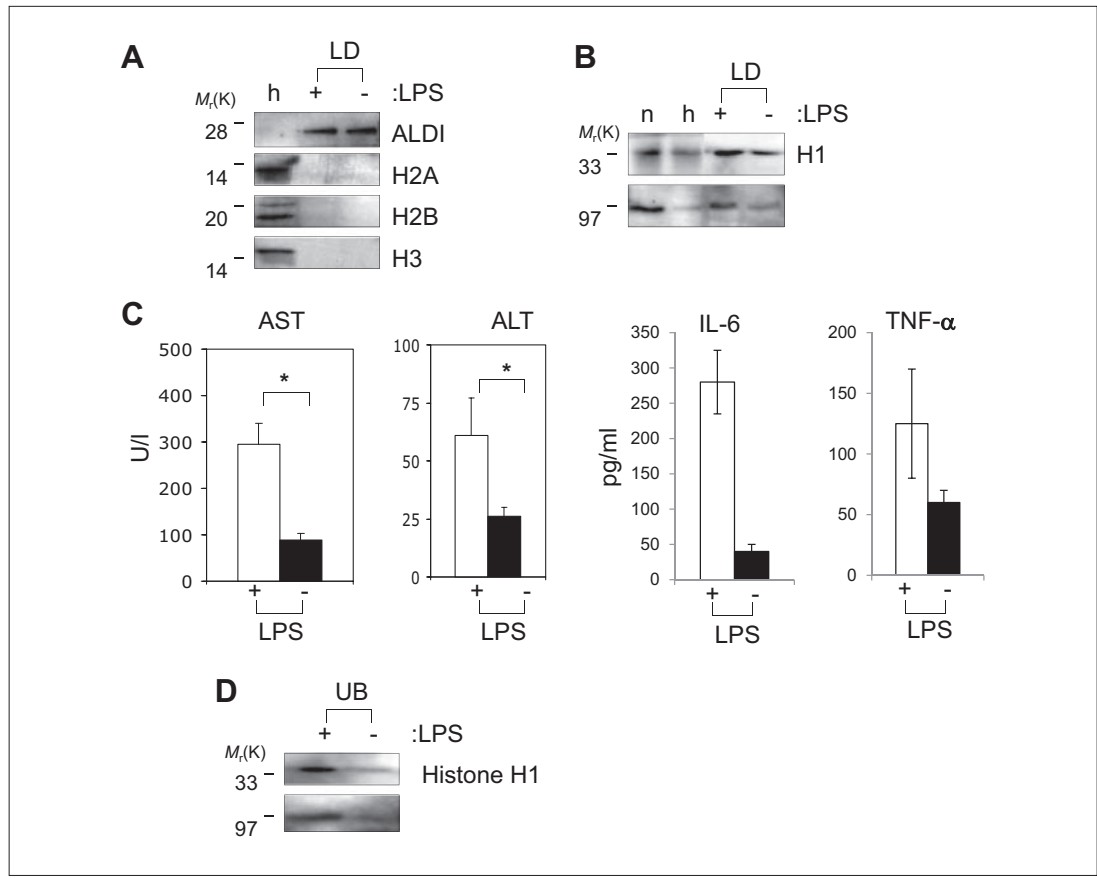

**Figure 6**. Histones are on mammalian LDs and respond to LPS. (**A**). Western blot analysis of LDs (LD) purified from hepatocytes of mice injected with (+) or without (−) LPS. Antibodies against ALDI, histones H2A, H2B and H3 were used. Whole liver homogenate (h) was used for comparison and as a control. (**B**) The presence of histone H1 (H1) on LDs (LD) purified from hepatocytes of mice injected with (+) or without (−) LPS was detected by immunoblot, and more H1 was present on droplets purified from LPS-treated animals. Equal total proteins from the nuclear fraction (n) and from whole liver homogenate (h) were used as comparison. (**C**). Mice were injected intraperitoneally with (+) or without (−) LPS, and transaminase levels (AST and ALT) and cytokine levels (IL-6 and TNFα) were quantified in units/l or units/ml in the serum; asterisk indicates statistical significance (p=0.05), confirming that LPS injection provoked the expected biological response. (**D**). Western blot analysis of histone H1 released into the buffer (UB) when purified LDs, from the liver of infection induced mice, were treated with LPS (+). Histone H1 is either minimally detected or not at all detected in the buffer in the absence of LPS (−). The band at 97 kDa in **C** and **D** represents histone H1 oligomers.

propose here that by using histone-droplet sequestration, cells can position histones in the cytosol, so that they are both available to interact with cytosolic bacteria, and also can evade constitutive pathways responsible for degrading free histones. Determining whether the murine droplet-bound histones actively contribute to host defense remains for future studies.

In principle, any kind of cell has the ability to form LDs, and thus employing droplet-bound histones as antibacterial defence might be quite general. Indeed, proteomic studies have revealed histones on LDs in tissues from larval and adult insects as well as in a range of mammalian cells (*Smolenski et al., 2007*; *Wan et al., 2007*; *Yang et al., 2010*; *Zhang et al., 2011*; *Larsson et al., 2012*), including mouse liver (*Figure 6*). Such a role of droplets in the antibacterial response may in part explain why the number and size of LDs increase in disease, from osteoarthritis to liver degeneration and cartilage overproliferation (*Tilney and Portnoy, 1989*; *Bielecki et al., 1990*; *Lowder et al., 2000*; *Gunjan and Verreault, 2003*; *Figueredo et al., 2009*; *Yang et al., 2010*). In mammalian leukocytes and macrophages, LDs have already been suggested to participate in the regulation of the host response to infection, by modulating the production of inflammation mediators like

eicosanoids (*Bozza and Viola, 2010*). Our result discloses an additional role of droplets in innate immunity, a role potentially conserved from flies to mammals. Since many medically relevant bacterial pathogens enter the cytosol (e.g. *Listeria monocytogenes*, *Shigella flexneri*, *Burkholderia pseudomallei*, *Francisella tularensis* and *Rickettsia* spp; *Ray et al., 2009*) and histones are also reported to have anti-fungal properties (*De Lucca et al., 2011*), and many such pathogenic fungi also enter the cytosol (e.g., *Cryptococcus neoformans*; *Bliska and Casadevall, 2009*), it may be that the system described here has surprisingly wide utility.

At least in the case of bacteria, histone release from the droplets is triggered by bacterial cell envelope components. This release likely allows an appropriate cellular response: freeing enough histones to kill bacteria, but minimizing problems (*Gunjan and Verreault, 2003*) of excess free histones interfering with endogenous cellular processes. We speculate that this release is achieved via direct binding between LPS and histones, as histones can indeed directly bind LPS (*Bolton and Perry, 1997*; *Augusto et al., 2003*) and the presence of LPS did not significantly affect the antibacterial activity of histones in the plate assay. We propose that the negatively charged LPS or LTA neutralize the positively charged histones and therefore weaken their electrostatic interactions with *Jabba* on the LDs; histone binding to droplets is charge-sensitive, as indicated by the ability of $CaCO_3$ to detach the histones from the droplets.

If histones are so effective, why load them not at high levels onto all LDs? The observation that droplet-bound H1 increases in mouse liver in response to a simulated bacterial infection might suggest that there is a cost to storing histones on droplets. Cells likely have to balance immediate histone availability with possible undesirable uncontrolled histone release from droplets due to metabolic consumption of the underlying droplets, or alternatively, potential secondary effects due to the excess histones causing saturation of histone-modifying enzymes (*Singh et al., 2010*).

## Materials and methods

### Isolation of LDs and treatments

*Drosophila* embryos (Oregon-R strain) were collected, aged and dechorionated with 50% bleach. LDs were purified as previously described (*Cermelli et al., 2006*). Briefly, LDs were isolated from total embryo lysates by sucrose gradient ultracentrifugation and solubilized in NP40 lysis buffer (10 mM Tris–HCl, pH 7.4, 0.5 mM EDTA, 1% NP40). Protein concentration was determined by Bradford dye-binding assay (Sigma-Aldrich, MO, USA). In some cases, the isolation of LDs was performed in the presence of 100 mM sodium/calcium carbonate, according to *Brasaemle et al., (2004)*, to remove proteins bound via electrostatic interactions. The presence of proteins histones H2A and H2B, and kinesin heavy chain in droplets was monitored using immunoblot analysis with anti-H2A (*Leach et al., 2000*), anti-H2B (Upstate Biotechnologies, Lake Placid, NY, USA), and anti-Khc (Cytoskeleton, CO, USA), respectively.

### Immunoblot analysis

Proteins were separated by SDS-PAGE and electro transferred to PVDF/nitrocellulose membranes. The membranes were blocked with 5% non-fat milk or 5% bovine serum albumin for an hour and then incubated with appropriate primary antibodies at desired concentrations for 1 hr at room temperature or over night at 4°C. Peroxidase conjugated donkey anti-rabbit/goat anti-mouse (1:10,000; Jackson ImmunoReseach) were used as secondary antibodies and the signals were monitored by Novex ECL kit (Invitrogen, CA, USA).

### Immunofluorescence

Embryos were centrifuged, heat-fixed, and stained with anti H2A and H2B antibodies (1:2000) as described (*Cermelli et al., 2006*). H2Av-GFP expressing embryos were imaged live without any fixation. Micrographs were acquired on a Leica SP5 confocal microscope or a Nikon Eclipse E600 fluorescence microscope with a 4 MP Spot Insight camera. Images were processed in Adobe Photoshop and assembled with Adobe Illustrator. In order to visualize the GFP bacteria inside microinjected embryos, a LSM710 confocal microscope was used.

### Antibacterial assays

To evaluate the antibacterial property of purified LDs/LD components, we performed colony forming assays, gel overlay assays as well as disc diffusion assays.

## The colony forming units (CFU) assay

The antibacterial activity of LDs was evaluated using a CFU assay as described previously (*Figueredo et al., 2009*) but with a slight modification of the conditions. Bacterial strains growing exponentially in Trypticase soy broth (TSB) at 37°C were collected by centrifugation, washed, and resuspended in Tris buffer (pH 7.4). Bacteria ($5 \times 10^3$ colony-forming units/ml) were incubated with LDs (non-treated or pre-treated) with a mixture of anti-histone antibodies (equal amounts of anti histone antibodies, at concentrations of 200 µg/ml, H1, H2A, H2B, H4 [all from Santa Cruz Biotechnology, CA, USA] and H3 [from Sigma-Aldrich, MO, USA] were mixed and treated with LDs [equivalent to ~500 µg total proteins] in a 1:100 antibody mix:LD ratio for an hour at 37°C) for 24 hr at 37°C in 3× TSB/LB. Samples were diluted with Tris buffer (pH 7.4), and various dilutions of the samples were plated on Trypticase soy-agar plates. Surviving bacteria were quantitated as CFU/ml on plates after incubation at 37°C for 12–18 hr.

The bactericidal activities of commercial pan-histone (Sigma-Aldrich, MO, USA) and AU-gel extracted histones from LDs sample were tested in vitro against *E. coli* ML35 as described previously (*Figueredo et al., 2009*). Bacteria growing exponentially in Trypticase soy broth (TSB) at 37°C were collected by centrifugation, washed, and resuspended in 10 mM PIPES (pH 7.4) supplemented with 0.01 volume (1% vol/vol) TSB (PIPES-TSB). Bacteria ($5 \times 10^6$ colony-forming units [CFU]/ml) were incubated with pan-histones (0–10 µg/ml) or extracted LD-histones (0–2 µg/ml) for 1 hr at 37°C in 50 µl of PIPES-TSB. Samples were diluted 1:100 with 10 mM PIPES (pH 7.4) and 50 µl of the diluted samples were plated on Trypticase soy-agar plates using an Autoplate 4000 (Spiral Biotech Inc., Bethesda, MD). Surviving bacteria were quantitated as CFU/ml on plates after incubation at 37°C for 12–18 hr, and data were analyzed and plotted using SigmaPlot (Systat Software, Inc., San Jose, CA).

## Gel overlay assay

The antimicrobial activity of purified LDs was examined by gel overlay assay as described by *Lehrer et al. (1991)*. Briefly, isolated LDs were subjected to acid-urea gel electrophoresis (AU-gel). Next the gels were washed in 10 mM sodium phosphate buffer, pH 7.4 (NAPB) and incubated at 37°C on top of 1-mm-thick underlay agar containing mid logarithmic-phase bacteria (*Escherichia coli*, DH5α strain). The underlay agar consisted of 10 mM NAPB, 1% vol/vol trypticase soy broth (TSB), 1% wt/vol of low-electroendosmosis-type agarose and 0.02% vol/vol Tween 20. After allowing 3 hr for diffusion of its proteins into the underlayer, the PAGE gel was removed and a nutrient-rich top agar was added on top of the underlay agar; the plate was then incubated for 18 hr at 37°C to allow growth of the bacteria. The location of antimicrobial polypeptides in the gels is revealed by a growth inhibition zone.

## Disc diffusion assay

An overnight broth culture (200 µl of *Escherichia coli*, DH5α strain) was spread over the surface of dried agar plates using a sterile glass spreader and allowed to absorb in the agar for 10 min. The plates were dried, inverted, at 37°C for approximately 30 min until the bacterial overlay had dried. Isolated LDs were resuspended in Top solution (25mM Tris-HCl, 1 mM EDTA, 1 mM EGTA) and pipetted (10 µl) onto a 7-mm sterile filter discs and the discs placed onto the agar plate and incubated at 37°C for 24 hr. For controls, 10 µl of Top Solution and antibiotics (Kanamycin 50 µg/ml) were added to the disc.

## Protein extraction from acid-urea gels

Duplicate samples of proteins from purified LDs were separated by AU-gel electrophoresis. One lane was stained by Coomassie Blue and used to identify the approximate position of histones in the unstained lane. This region of the unstained lane was cut out, macerated, and incubated overnight in 5% acetic acid. The solution with extracted proteins was lyophilized, and the pellet resuspended in 1% TSB.

## In gel digestion

LDs proteins were separated by 1-D SDS-PAGE and stained by Coomassie Blue. Proteins bands were excised, in-gel digested by trypsin, extracted, and concentrated for LC-MS/MS analysis as described (*Huang et al., 2001*).

## Mass spectrometry analysis by LC-MS/MS

The tryptic digests were analyzed by LC-MS/MS using a nanoLC system (Eksigent, Inc., MA, USA) coupled with Linear Ion Trap (LTQ)-Orbitrap XL mass spectrometer (Thermo-Electron Corp, OH, USA).

The LC analysis was performed using a capillary column (100 μm i.d. × 150 mm length) packed with C18 resins (GL Sciences, CA, USA) and the peptides were eluted using a linear gradient of 2–35% B in 105 min; (solvent A, 100% $H_2O$/0.1% formic acid; solvent B, 100% acetonitrile/0.1% formic acid). A cycle of one full FT scan mass spectrum (350–1800 $m/z$, resolution of 60,000 at $m/z$ 400) was followed by 10 data-dependent MS/MS acquired in the linear ion trap with normalized collision energy (setting of 35%). Target ions selected for MS/MS were dynamically excluded for 30 s.

### Drosophila strains

We used Oregon-R as our wild-type stock. We employed two mutant alleles of *Jabba* (also known as CG42351): *Jabba*[zl01] is a promoter deletion and expresses no *Jabba* protein in early embryos; *Jabba*[f07560] is a transposable element insertion between two coding exons of *Jabba*, resulting in a severely truncated *Jabba* protein. A comprehensive molecular and phenotypic description of *Jabba* mutant alleles will be published elsewhere (*Li et al., 2012*). Alleles *Jabba*[zl01] and *Jabba*[f07560] were derived independently, and thus likely share few, if any, unknown secondary mutations. Both alleles eliminate droplet-bound histones (*Figure 3D*, *Figure 3—figure supplements 1–3*).

### Embryo bacterial injections

Cultures of the bacterial strain of interest (*E. coli DH5α*, *S. epidermidis*, *L. monocytogenes*, *B. subtilis* [*hly*] and *YD133-GFP*; an *E. coli* K-12 bacterial strain expressing GFP) were grown to the log phase. An appropriate volume of the bacterial suspension was pelletted, washed with PBS and re-suspended in injection buffer (5 mM KCl, 0.1 mM sodium phosphate pH 6.8, 5% [vol/vol] of McCormick green food color). Precellular blastoderm stage embryos were injected manually using a Narishige IM300 microinjector. Injected embryos were maintained at 25°C in a fly incubator and the survival was monitored daily until they hatched into larvae. Percentage survival was normalized with respect to the survival of embryos injected with buffer only.

An estimate of the bacterial load in the GFP-bacteria injected cases was done by image analysis. We searched in each embryo for the region(s) with the most bacteria visible, and quantified that number, using images with same field of view. At least three different embryos were used in each group. At the same time, we noted how many such high-bacteria fields were typically present in an embryo, as well as typical bacterial counts in the other fields. We estimate the embryo can be covered by approximately 10 independent fields of 40 μ each. For the wild-type, at t = 2 hr, the average number of bacteria in the (maximal) field of view was 24.3 ± 6.4, and there were between three and four such fields per embryo, with few bacteria elsewhere, leading to an estimate of ~84 bacteria per embryo. For the *Jabba* mutant embryo, the average number of bacteria in the (maximal) field of view was 23.3 ± 3.2, and there were between three and four such fields per embryo, with few bacteria elsewhere, leading to an estimate of ~79 bacteria per embryo. For the t = 24 wt embryos, the average maximal field had 21.7 ± 9.2 bacteria, but there was only one such field per embryo, typically with 3–4 other areas with 1–5 bacteria per field, resulting in an estimate of 33 bacteria per embryo. For the t = 24 *Jabba* mutant embryos, approximately 1/3 of the fields had the high bacterial count (~180 ± 16.1 bacteria), and the rest typically had 10–30 bacteria per field, leading to an estimate of 840 bacteria per embryo. Finally, for the t = 48 hr wild-type, at most only one or two fields had any detectable bacteria (some embryos had none) so we estimate between 2–6 bacteria per embryo. In contrast, for the t = 48 *Jabba* embryo, every field was full of bacteria (200–250 bacteria per field), so assuming 10 such fields we estimate 2000–2500 bacteria per embryo.

### Adult fly bacterial infections

Adult flies (both wild type and *Jabba* mutant, 2–3 days old) were subjected to an infection assay. The flies were anesthetized with $CO_2$, and pricked under the wing with a fine metallic needle (33G) which was first dipped into either buffer or a suspension of the bacteria *Listeria monocytogene*. To make the bacterial suspension, a dilute overnight culture was grown to an absorbance of 0.5–0.6 at wavelength 595 nm. From this log-phase bacterial suspension, 5 ml was pelletted and the pellet washed with PBS. This bacterial pellet was then suspended in 200 μl of PBS and the metallic needle was dipped in for the bacterial infection assays. After the infection, the flies were maintained in 25°C fly incubator. Every 24 hr, the number of dead and live flies was counted. The percentage of survival calculated from these counts was then plotted against days post injection as shown in *Figure 4*. Approximately 10 live flies from the both wild type and *Jabba* mutant at days 0 and 3 were sacrificed and cytosolic extracts were

prepared using ready prep protein extraction kit (Bio-Rad Laboratories, Inc, CA, USA) according to the manufacturers' protocol. Equal amounts of cytosolic extracts were then plated on agar plates to estimate the relative viable cytosolic bacterial load as shown in *Figure 4C*. The quantification of band intensity in the *Figure 4C* was done using Image J software.

## Mice injection and hepatic lipid droplet treatment

C57BL/6 mice were kept under a controlled humidity and lighting schedule with a 12-hr dark period. All animals received human care in compliance with institutional guidelines regulated by the European Community. Food and water were available ad libitum. Male mice of approximately 8–12 weeks old were intraperitoneally injected with 20 ng LPS diluted in NaCl 0.9% or with an equivalent volume of NaCl and fasted for 16 hr. Hepatic LDs were isolated exactly as described in *Turro et al., (2006)* (*Wan et al., 2007*). The protein fraction of LDs was separated by precipitation with 33% of cold TCA and the resulting pellets resuspended in 50 µl of 250 mM Tris, 2% SDS. The protein content was determined by the method of Lowry, and equal amount of protein of LDs was separated by SDS-PAGE. Western blotting was performed as described previously. In some experiments, the LDs obtained from LPS-treated mice were diluted 1:3 in a 250-mM Tris buffer and incubated with 1 mg/ml of LPS or an equivalent volume of NaCl for 2 hr at 4°C. A soluble fraction of histones and other proteins was separated from the intact LDs by centrifugation at $16,000 \times g$ for 15 min in an Eppendorf microfuge. Finally, equal volumes of soluble fractions were separated by SDS-PAGE and the presence of H1 determined by Western blotting. The primary antibodies used in immunoblot analysis of mouse samples were anti-histone H1, clone AE-4 from UPSTATE biotechnology (Millipore, MA, USA), Anti histone H1: sc-10806, anti histone H3 (FL-136): sc-10809 and anti histone H2B (FL-126): sc-10808 (from Santa Cruz Biotechnology, CA, USA), anti histone H1.2, anti histone H2A and anti histone H2B (from Abcam, Cambridge, UK). The polyclonal anti-ALDI used in this study is described in *Turro et al., 2006*.

## Acknowledgements

We thank the Huang, Warrior, Arora and the Ouellette laboratories members for their help during this study, especially Jennifer Mastroianni and Dr. Doug Bornemann for technical support. We thank Dr. Daniel A. Portnoy, UC Berkeley and Dr. Teuta Pilizota, Princeton University for kindly providing the bacterial strains. We also thank Samuel Martinez Gross for his work in quantifying bacterial counts in the fluorescent images.

## Additional information

### Funding

| Funder | Grant reference number | Author |
| --- | --- | --- |
| National Institutes of Health | GM64624 | Steven P Gross |
| National Institutes of Health | GM64687 | Michael A Welte |
| National Institutes of Health | AG031531 | Michael A Welte |
| Spanish Ministerio de Ciencia e Innovación | BFU2011-23745 | Albert Pol |
| National Science Foundation | LifeChips-IGERT fellowship | Silvia Cermelli |

The funders had no role in study design, data collection and interpretation, or the decision to submit the work for publication.

### Author contributions

PA, Conception and design, Acquisition of data, Analysis and interpretation of data, Drafting or revising the article; SC, Conception and design, Acquisition of data, Analysis and interpretation of data, Drafting or revising the article; ZL, Acquisition of data, Analysis and interpretation of data, Contributed unpublished essential data or reagents; AK, Acquisition of data, Analysis and interpretation of data; MB, Conception and design, Acquisition of data, Analysis and interpretation of data; RS, Acquisition of data; LH, Analysis and interpretation of data; AJO, Conception and design, Analysis and interpretation of data, Drafting or revising the article; AP, Conception and design, Acquisition of data, Analysis and

interpretation of data, Drafting or revising the article; MAW, Conception and design, Analysis and interpretation of data, Drafting or revising the article; Contributed unpublished essential data or reagents; SPG, Conception and design, Analysis and interpretation of data, Drafting or revising the article.

### Ethics

Animal experimentation: All animals received human care and experimental treatment authorized by the Animal Experimentation Ethics Committee (CEEA) of the University of Barcelona (expedient number 78/05), in compliance with institutional guidelines regulated by the European Community.

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
