## [Author Response]

*An important point that is currently not discussed in the paper is that, as of now, this phenomenon has been shown to apply to fly embryos (with some interesting results in murine liver added). But we feel there should be some discussion of whether this might apply to the adult organism – perhaps even some presentation as to whether adult Jabba flies are compromised or not when confronted with bacterial pathogens, if such trials have been done. While such experiments are not absolutely required (eLife policy is to suggest just the absolutely essential experiments), the text must be modified to bring forth this important limitation; as of now the effect has been seen only in early embryos and not in adult flies*.

We agree that this was an interesting and important question. In the new version of the manuscript, we answer it in the affirmative—Figure 4 is entirely new, and shows new data quantifying survival of adult flies challenged with bacteria. Further, Figure 4 shows that we can indeed detect histones in the adult cytosol, and that histone levels are decreased in the Jabba mutant background.

*We recommend that the effect of Jabba be tested in another mutant background (jabba/deficiency) to exclude a possible background effect*.

In fact, this was not an issue—the experiments presented had been done in two entirely different genetic backgrounds, but our description was lacking. We have now modified the text to show the exact details of the two different mutants used in this study to address the background effect. Alleles *Jabba*^*zl01*^ and *Jabba*^*f07560*^ were derived independently: *Jabba*^*zl01*^ is due to imprecise excision of a P element inserted near the Jabba promoter. *Jabba*^*f07560*^ is due to the insertion of a PBac element in the middle of the Jabba coding region. As the two alleles were generated in entirely different genetic backgrounds, this rules out any genetic background effects. This has now been clarified in the text.

*There were also some issues raised regarding the experiments using mammalian cell lines. Firstly, in the context of immunity, measuring the efficiency of an LPS infection is usually defined by cytokine response rather than hepatic injury. Secondly, if histones do act as cytosolic antibacterial agents in mammalian cells, it might do to include a bacterial killing assay with intracellular bacteria and mammalian cells. In short, it would be good to tone down this section as well because the experiments presented are also rather limited, if provocative*.

The cytokine levels in the murine study are now included in the revised version of the manuscript (see Figure 6). We have not been able to do a bacterial killing assay with the murine droplets, because a great deal of research in Spain (where our collaborators are located) is currently on hold due to the financial crisis there; the mouse facility is closed for a month or two. Given our new data on whole-fly data, and the expected large delay in being able to carry out the murine-droplet killing assays, we propose to leave establishing the efficacy of the murine droplets for future work. Accordingly, this section has been toned down as suggested by the reviewers, and in particular, we now include the sentence: “Determining whether the murine droplet-bound histones actively contribute to host defense remains for future studies.”